# Oral Gels as an Alternative to Liquid Pediatric Suspensions Compounded from Commercial Tablets

**DOI:** 10.3390/pharmaceutics16091229

**Published:** 2024-09-20

**Authors:** Monika Trofimiuk, Małgorzata Sznitowska, Katarzyna Winnicka

**Affiliations:** 1Department of Clinical Pharmacy, Medical University of Bialystok, 15-222 Bialystok, Poland; monika.trofimiuk@umb.edu.pl; 2Department of Pharmaceutical Technology, Medical University of Bialystok, 15-222 Bialystok, Poland; katarzyna.winnicka@umb.edu.pl; 3Department of Pharmaceutical Technology, Faculty of Pharmacy, Medical University of Gdansk, 80-210 Gdansk, Poland

**Keywords:** valsartan, candesartan cilexetil, carbomer, cellulose polymers, pediatric oral gels, stability

## Abstract

The aim of the study was to propose pharmacy-compounded oral gels as a new and alternative dosage form that is attractive to children as having a better masking taste than syrups and reducing the risk of spilling. The application and physical properties of the gels prepared with cellulose derivatives (hydroxyethylcellulose and carmellose sodium) or carbomers were evaluated. The results of the study showed the most suitable consistency, viscosity, and organoleptic properties for gels prepared with carbomer and cellulose derivatives at concentrations of 0.75% and 2.0%, respectively. The microbial stability of the gels was guaranteed by the use of methylparaben and potassium sorbate. VAL (valsartan) and CC (candesartan cilexetil) tablets, often used off-label in children, were pulverized and suspended in the hydrogel bases, resulting in final drug concentrations of 4 mg/g and 1 mg/g, respectively. There was no significant change in viscosity and consistency parameters when the pulverized tablets were added, and only small changes in viscosity and consistency were observed during 35 days of storage, especially in the gels with sodium carmellose and candesartan. On the basis of the drug assay, an expiry date of 25 °C was recommended: 35 days for valsartan and 14 days for candesartan preparations.

## 1. Introduction

The lack of age-appropriate dosage forms with a dose easily adjustable is still a highlighted problem in pediatric therapy [1,2,3,4,5]. The common practice is to prescribe medicinal products that are in tablet or capsule form and have a license for use only in adults [6,7]. Preparation of pediatric dosage forms from such products is an off-label use that is often practiced clinically. To adjust the dose in such situations, pharmacists compound powders in capsules (gelatine or starch), or in paper sachets, or prepare syrups. Powders, before administration to small children, are sprinkled and mixed with soft food, milk, or another beverage. Such manipulation brings the risk of drug–food interaction or administration of an incomplete dose (loss of powder) while emptying the capsule [8,9,10,11,12]. Oral liquid extemporaneous dosage forms (syrups, suspensions, and solutions) eliminate this kind of problem; however, palatability is still a challenge [12,13,14,15,16,17,18,19,20]. Although liquids are a popular alternative to powders in the USA, they are less frequently or rarely compounded in European countries [11,12,13,14,15,16,17,18,19,20,21].

Whereas pediatric topical oral gels (for example, to treat inflammation or infection in the oral cavity) are widely available, oral gels are less common as alternatives to tablets or syrups [22,23,24]. Commercial oral pediatric hydrogel dosage forms registered as medicinal products have already been introduced as attractive alternatives for dietary supplements with vitamins. Like any soft food, semisolid oral hydrogels may assess ease of swallowing without mixing powders with food, which always creates the risk of incompatibility. Thus, oral administration of hydrogels is a new trend in pediatric therapy and brings positive recommendations with respect to safety, good palatability, and flexibility of dosage by measured volume. Hydrogels could be an attractive dosage form because of their smooth structure, which improves the child’s feelings during administration, better taste masking, and less risk of liquid spilling from the spoon. They may be easily packed into single-dose sachets or in tubes. A dose from the latter may be measured using spoons or oral syringes.

If the active substance is suspended in the gel, the homogeneity during the whole time of storage should be assured by the proper viscosity of the carrier because, in contrast to liquid suspensions, the sediment cannot be restored by shaking. There is some risk that the high viscosity of the formulation may reduce the bioavailability of the drug or delay its absorption, but the same risk exists when mixing a child’s medicine with food. Although there is a lack of research on this topic, it is definitely possible to consider it, especially when drug absorption is rapid. On the other hand, absorption from a tablet is determined by its disintegration, while in a gel, the medicinal substance is in the form of already small particles.

Valsartan (VAL) and candesartan cilexetil (CC) are used in the treatment of hypertension and belong to the therapeutic category of angiotensin II receptor blockers. Commercially available are tablets with CC (4, 8, 16, and 32 mg—Atacand and generic products). CC is slowly absorbed from the gastrointestinal tract, with a t_max_ of 3–4 h [25,26]. The activity, safety, and pharmacokinetics data of CC were studied in children from 1 to 17 years of age, with a dose for children between 1 and 6 years of 0.2 mg/kg/day (the usual dose ranges from 0.05 to 0.4 mg/kg/day) and from 6 years of age and older (weight < 50 kg) of 4–8 mg once daily (the usual dose ranges from 2 to 16 mg/day) [25,26,27,28,29,30,31]. According to the Atacand FDA leaflet, preparation of oral suspension by dispersing tablets in commercially available suspending media (Ora-Sweet SF, Ora-Plus, or Ora-Blend) is recommended [25]. The physicochemical properties of CC and VAL are presented in Table 1.

Tablets with VAL (40, 80, 160, and 320 mg—Diovan and generic products) are indicated for adults and children 6–16 years old [36,37,38,39]. For younger children from 1 year old, the Diovan solution is recommended with a starting dose of 1 mg/kg up to a maximum dose of 4 mg/kg once daily (with dose adjustment based on blood pressure response and tolerability). For children older than 6 years of age and < 35 kg weight, 20 mg once daily (7 mL solution) is recommended, and for > 35 kg weight, 40 mg (14 mL solution) is advised [40]. Absorption from the solution is faster than from tablets (t_max_ is 1–2 h for a solution and 2–4 h for tablets). The systemic exposure to valsartan (AUC) is 60% higher with the suspension or solution compared to tablets, and when switching between forms, the dose of VAL may need to be adjusted [40,41]. Unfortunately, the availability of this product in the form of a solution is very limited in many European countries, and the off-label use of VAL in tablets is very common.

Suspension-type syrups compounded from Diovan tablets (VAL concentration 4 mg/mL) and Ora-type media can be stored in a glass bottle for either up to 90 days at temperatures below 30 °C or up to 75 days at refrigerated conditions [41]. Syrups prepared with Atacand tablets, with CC concentrations of 1 mg/mL or 2 mg/mL, should be used within 30 days (up to 100 days when unopened) when stored at room temperature [25,42,43,44]. For simpler suspending media, based on sucrose, stability of the suspensions for 3 or 4 weeks at room temperature was declared [21].

The aim of the current study was to develop hydrogels as alternative formulations to VAL and CC liquid suspensions. The new compounded formulations are proposed for easier drug administration, as described above. The chemical stability of the drugs in the gel vehicles was evaluated by using a HPLC method. Physical stability and the rheological properties of the final formulations were investigated at different temperatures during the time. The results obtained should allow gel vehicles to be considered a new dispersing medium for active substances, including when the drugs are obtained by manipulating tablets or capsules. The proposed extemporaneous formulations could also represent a valuable alternative to Ora^®^ products when they are not available on the market.

The study did not include the impact on the bioavailability of the drug in terms of both crushing tablets and incorporation in a viscous medium. The tested active substances belong to BCS class II and are sparingly soluble in water, and the risk of changing bioavailability should be taken into consideration. However, a similar risk occurs when combining medicines with food, which is a common practice in administering drugs to children.

## 2. Materials and Methods

### 2.1. Materials

Hydroxyethylcellulose (HEC) (Natrosol 250 HR was purchased from A.C.E.F., Piacenza, Italy), sodium salt of carmellose (high-viscosity carboxymethylcellulose, CMC) (Sigma Aldrich, St. Louis, MO, USA), and carbomer (CAR) dedicated to oral use (Carbopol 974 P NF, Noveon, Cleveland, OH, USA) were used as gelling agents. Other chemical substances and solvents used in hydrogels were of pharmacopoeial quality: methylparaben (Fluka, Steinheim, Germany), glycerol (Amara, Krakow, Poland), potassium sorbate, and sorbitol (POCH, Avantor Performance Materials Poland, Gliwice, Poland).

Oral pediatric hydrogels were prepared with tablets obtained from a local pharmacy. For VAL hydrogels, Diovan^®^ (Novartis, Nürnberg, Germany) coated tablets were used at a strength of 160 mg. The tablets contain the following excipients: microcrystalline cellulose, polyvinylpyrrolidone, colloidal silica, magnesium stearate, hypromellose, titanium dioxide, polyethylene glycol 8000, and iron oxides. For CC preparation, Atacand^®^ (Astra Zeneca, Södertälje, Sweden) uncoated tablets with a strength of 16 mg were used, composed of the following excipients: calcium carboxymethylcellulose, hydroxypropylcellulose, lactose monohydrate, magnesium stearate, corn starch, polyethylene glycol 8000, and iron oxides.

Reagents used for high-performed liquid chromatography (HPLC) analysis, that is, acetonitrile (POCH, Gliwice, Poland), methanol (J.T.Baker, Deventer, Netherlands), and ortho-phosphoric acid (Fluka, Steinheim, Switzerland), were of HPLC grade. The VAL and CC substances used for the calibration curve were donated by Polpharma (Starogard, Poland).

### 2.2. Preparation of Hydrogels

The composition of the placebo hydrogels is shown in Table 2. Methylparaben and potassium sorbate were used as preservatives. To increase palatability, sucrose, sorbitol, and glycerol were used as sweetening agents. Additionally, glycerol was added as a humectant. Hydrogel media were compounded by dissolving the following excipients in purified water: methylparaben, potassium sorbate, sorbitol, and sucrose. Afterward, glycerol was added, and using a mechanical stirrer, the polymer was dispersed and dissolved to form a hydrogel.

In the case of CAR, after the polymer was dispersed, neutralization with a 20% solution of sodium hydroxide (6.1 M) was performed to produce a hydrogel [45,46].

VAL and CC were incorporated into hydrogel carriers at concentrations of 4 mg/g and 1 mg/g, respectively. The following is required for 100 g of the gel: 2.5 tablets of Diovan containing 160 mg of VAL and 6.25 tablets of Atacand with a dose of CC 16 mg. Hydrogel formulations were prepared by mixing the pulverized tablets (787 mg of Diovan and 777 mg of Atacand) with the hydrogel medium. The suspension was prepared in a mortar by trituration of the powder with portions of the hydrogel media. The oral hydrogels were stored in plastic containers.

### 2.3. Short-Term Stability Studies

The short-term stability studies were carried out for the preparations stored at 25 °C and 4 °C for a period of up to 35 days. Every 7 days, the gels were analyzed by visual and microscopic observations, pH, viscosity, and other mechanical parameters were measured, and VAL and CC content was assayed. Microscopic observation was carried out using an optical microscope (Moticon, Wetzlar, Germany) with a camera (Motic BA 400).

### 2.4. Analysis of Viscosity, Rheology, and Texture of Oral Hydrogels

A cone/plate viscometer was employed (Brookfield RVDV-III Ultra Rheometer with Rheocalc software, manufactured by Brookfield Engineering Lab., Middleborough, MA, USA), and the dynamic viscosity of the hydrogels was measured as a relationship between shearing stress and shear rate. Rheological behavior was evaluated on the basis of a flow curve (a plot of viscosity to shear rate) and a hysteresis loop (a plot of shear stress to shear rate) [47,48]. The measurements were carried out at a temperature of 25.0 °C ± 0.1 °C controlled with a thermostatic water bath.

Additionally, some other mechanical parameters of the hydrogels were measured using a TA.XT Plus Texture Analyzer (Stable Micro Systems, Godalming, UK) equipped with A/BE Back Extrusion Ring (Ø 40 mm) and Texture Exponent software [49]. The following operating parameters were set: Pre-test speed of a probe 1.5 mm/s, test and post-test speed 2.0 mm/s, penetration distance of the probe in the sample was 5 mm, and trigger force of the probe was 10 g. The relationship between time and measured force was obtained, and the example curve is presented in Figure 1. The following mechanical parameters were measured: firmness [g], minimum adhesive force [g], adhesiveness [g∙s], and cohesiveness [g∙s] [48,49,50,51,52,53,54]. The analysis was performed at an ambient temperature. Each sample was measured in triplicate with empirically established 1-h time intervals between individual measurements (time necessary for hydrogel structure reconstitution).

### 2.5. HPLC Analysis of the Drug Content

To determine the VAL and CC content in the prepared oral hydrogels, HPLC was carried out using an Agilent HPLC system (Agilent Technologies, Waldbronn, Germany). Previously described and validated methods were employed [21,55,56,57,58,59,60]. The specificity and accuracy of the method for the analysis of oral hydrogels were confirmed. An isocratic separation on an Eclipse XDB—C18 column (4.6 × 150 mm, 5 µm) at 25 °C was performed with mobile phases: a mixture of a phosphate buffer pH 2.7 (0.065 M) with acetonitrile (55:45; *v*/*v*) for VAL and a mixture of a phosphate buffer pH 3.0 (0.05 M), methanol, and acetonitrile (30:20:50; *v*/*v*) for CC. The flow rate of the mobile phase was 1 mL/min, the UV detector was set at 215 nm, and the injected sample volumes were 20 µL. The retention times for VAL and CC were around 5 min and 13 min, respectively.

The three independent samples from each hydrogel were analyzed in triplicate. The 1.0 g sample of hydrogel was diluted to 10.0 mL with methanol. The solution was thoroughly shaken by hand for 5 min and then centrifuged for 15 min at 4000 rpm. The supernatant (1.0 mL) was diluted to 10.0 mL with a mobile phase to obtain solutions at concentrations of 10 µg/mL and 40 µg/mL for CC and VAL, respectively.

The concentration of the drugs in gels was calculated from calibration curves obtained from the analysis of standard solutions in mobile phases within the concentration range of 5–15 µg/mL and 10–80 µg/mL for CC and VAL, respectively. The linearity of the curve was confirmed by a correlation coefficient greater than 0.999. Interday precision for CAN and VAL was 4.52% and 1.15%, respectively (CV—coefficient of variation).

## 3. Results and Discussion

For the preparation of hydrogels, popular hydrophilic viscosity-increasing polymers were used, namely carbomer (CAR) and cellulose derivatives: hydroxyethylcellulose (HEC) and carmellose sodium (CMC). These polymers are excipients with wide uses and functions in the technology of pharmaceutical dosage forms (solid, semisolid, and liquid). They are commonly used in pediatric liquid dosage forms (syrups and suspensions) as thickening agents to prevent sedimentation of the dispersed phase. CMC is the most common, and CAR or HEC are used less frequently in liquid preparations for children, while they are very often applied as gelling agents in topical gels for children administered in oral cavities (pain relief, local anesthetics, and antifungal drugs). Based on adult safety and toxicity data, HEC, CMC, and CAR are considered safe for oral use in children, although only stronger data confirming the efficacy and safety in children have been published for CMC (STEP database) [61,62].

Among the three tested polymers, HEC is a non-ionic compound, while CAR and CMC are present in an ionic form, with pK_a_ 6.0 and 4.3, respectively [63,64,65,66]. The polymers used in the study are classified as high (CMC and HEC) or medium-high viscosity types gel-forming polymers.

The prepared hydrogels contained humectants (glycerol), sweeteners (sucrose and sorbitol), and preservatives. To ensure microbial stability, the addition of preservatives was necessary. The most frequently used are esters of p-hydroxybenzoic acid (parabens): methylparaben and propylparaben or potassium sorbate [67]. A combination of both was used based on previous findings that this combination was suitable for compounded syrups [21]. Sorbitol and glycerol concentrations are 4% and 5%, respectively, which is lower than in many syrups, and these compounds in such amounts should not be problematic regarding drug bioavailability [21,68,69,70].

The gels were prepared with VAL and CC at concentrations of 4 mg/mL and 1 mg/mL, as proposed for compounded syrups [18]. The concentration of crushed tablets in both gels was approximately 0.8%.

The density of the preparations is close to 1.0 g/mL (1.01–1.08 g/mL), which means that in practice, the dose measured in mass corresponds with the dose measured by volume. This allows for volume dose measurement using an oral syringe, which can also be a packaging container for the compounded gel. For children between 1 and 6 years, the doses are in the range of 3–6.5 mL for VAL and 0.5–4.0 mL for CC gels.

### 3.1. Visual and Microscopic Observations

The organoleptic assessment allowed us to propose optimum concentrations of the polymers in the gel. Depending on the polymer type, the following concentrations resulted in satisfying and visually similar consistency: 2.0% *w*/*w* for HEC and CMC and 0.75% *w*/*w* for CAR. The taste of the gels was pleasant.

The drug-containing gels prepared from tablets were suspension-type formulations that were non-transparent due to both insoluble active substances and excipients (silica, microcrystalline cellulose, magnesium stearate, and titanium dioxide). The gels prepared from the Atacand^®^ tablets were pink due to the tablet dye color (iron oxides). Figure 2 presents hydrogels prepared from VAL and CC tablets. Visual observation of the preparations showed no changes in color, odor, or taste upon 35 days of storage at a temperature of 25 °C or 4 °C.

VAL and CC belong to the II BCS class and have poor solubility in water: 0.18 mg/mL and 0.10 mg/mL, respectively [32,33]. Therefore, it may be assumed that they are present in the gels in the suspended phase, together with other insoluble tablet excipients.

Homogenous suspension-type formulations with all three hydrogel media were easily prepared using a classical method with manual trituration of pulverized tablets with the vehicle in a mortar. The coating in the Diovan tablets was easily pulverized, and there was no need to sieve it or remove it in any other way.

The size of the suspended tablet particles was 10–25 µm and was similar to the VAL and CC preparations. The microscopic analysis during the stability study period (35 days) showed no evidence of recrystallization, and no particles greater than 30 µm were observed.

### 3.2. pH Measurements

The pH of the drug-free hydrogels was 6.5, 7.1, and 6.3 for HEC, CMC, and CAR, respectively. After the addition of CC, a significant change in pH (*p* < 0.05) occurred only in the CAR gel, where the pH increased to 6.8. The addition of VAL tablet mass caused a decrease in pH value below 6.0, with a significant drop (*p* < 0.05) in VAL_HEC gel (pH 4.7) and VAL_CMC gel (pH 5.1). The pH of VAL dispersion in water is 4.0, so it may be assumed that the observed decrease in pH in VAL hydrogels is caused by VAL partially dissolving in hydrogel media. During storage, no change in pH was noted for VAL hydrogels, and only a small drop in pH (by 0.2–0.3 units) was observed in the CC formulations after 28 days, independent of storage temperature. Changes in pH values during storage were considered significant for a difference of ± 0.5 units from the initial value.

### 3.3. Viscosity, Rheology and Consistency

The physical properties of the formulations were evaluated as important features, not only for handling and easy administration of the gels but also for detecting any quality changes upon storage.

Viscosity and rheology, together with other mechanical properties, such as firmness, cohesiveness, and adhesiveness, should be taken into consideration if desirable application properties and patient acceptability of semiliquid or semisolid preparations are evaluated. These are also important factors that influence the accuracy of the dosage when spoons or syringes are used as dosing devices.

Figure 1 presents an exemplary graph obtained as a result of the gel analysis performed using a texture analyzer.

Figure 3 presents the mechanical parameters (absolute values) of the hydrogels, determined as described above. Despite the similar consistency of all gels evaluated visually, placebo gels prepared with HEC were characterized by the lowest firmness, cohesiveness, and adhesiveness, while CMC and CAR were similar in their physical texture parameters but with larger cohesiveness of CMC.

Results of the mechanical analysis showed that the addition of the VAL or CC tablets to HEC, CMC, and CAR hydrogel media generally resulted only in a rather slight increase in the majority of the mechanical parameters. Small but significant changes were noted in the CAR hydrogels: in the presence of VAL, firmness and adhesive force were reduced, and in the presence of CC, an increase in adhesiveness and cohesiveness was observed, but firmness was unchanged. Atacand^®^ tablets with CC contain calcium carboxymethylcellulose, and this excipient could be responsible for these changes because calcium ions might interact with ionic polymers, while HEC gels were insensitive [70].

Figure 3 also presents the characteristics of the preparations stored for up to 35 days at 25 °C. There was no significant effect on changes in mechanical parameters depending on the storage temperature (similar values for the samples at 4 °C were obtained). As a rule, an increase in the measured mechanical parameters is observed during storage; however, significantly larger changes were found in the gels with CC, especially in the CMC_CC. Similar results were observed by Chalah et al. [71], who noted a sharp increase in consistency for 2% CMC during storage. The observed time-related changes may have resulted from the influence of excipients included in the tablet mass (i.e., their slow dissolution and swelling). Very high values of firmness (807 g on the 35th day of storage) were also noted in the VAL_CMC gel, while the other parameters measured in this gel were stable. Observed changes did not influence; however, the application properties and organoleptic effects, such as thinning or thickening of gels after the addition of powdered tablets, were not noted.

The gels were also tested using a rheometer to determine their dynamic viscosity and rheology. It is difficult to present rheological measurements in a single number, as the graphs show a non-linear relationship between the response and the continuously changing mechanical forces applied. In Table 3, the values of viscosity recorded at the particular shear rate, that is, 8 s^−1^, are presented, and Figure 4 shows the hysteresis loops recorded as a relationship between shear stress and shear rate. It was shown that benzalkonium chloride or sodium benzoate, in high concentrations of 0.1% *w*/*v*, affects the rheological parameters in carbomer’s gel, reducing their viscosity and causing turbidity [70,72]. Considering that the addition of substances that prevent the growth of microorganisms is necessary, both the type and concentration of preservative added must be taken into account. No decrease in viscosity of the gels with carbomer and CC or VAL tablets was observed (Table 3).

The viscosity of the CMC gel measured with a rheometer was the highest, while HEC and CAR demonstrated similar viscosity (Table 3). The strongest thixotropic properties, measured by the size of the hysteresis loop, were also observed for the CMC gel (Figure 4). Such a difference was not observed when the measurements were performed with a texture analyzer (Figure 3). This means that mechanical measurements depend heavily on the measuring technique and the parameters determined. Finally, the results may be treated as complementary, and careful interpretation is required [48,49].

The solid phase of the VAL or CC tablets added to the gels resulted in only minor changes in their viscosity (Table 3). This could be expected, as the content of the solid particles in the gel was only 0.8%. After 35 days of storage, the changes observed in viscosity were not greater than 35%, which is not a large difference for this parameter.

No significant change in rheology was observed when CC was added to CAR gel, or VAL was mixed with CMC gel. In addition, only a small increase in shear stress was noted in the CMC gel with CC. On the other hand, in the case of both drugs, a significant reduction in shear stress was observed for the HEC gels, and this effect was also found when VAL was added to the CAR gel. Thixotropy (the area of the hysteresis loop) did not change. During storage for 35 days, a small increase in the gel structure, measured by shear stress, was noted only in the drug-loaded CMC gels, and this effect was not related either to the type of active substance added or to the temperature of storage (4 °C or 25 °C).

It may be concluded that changes in the physical characteristics of the gels in the presence of the added active substances in the form of the pulverized tablets (<1% content of solids) do not change the rheological behavior of the gels, or the change is too small to have an impact on the gel’s applicability.

### 3.4. Chemical Stability of VAL and CC in Oral Hydrogels

The exemplary chromatograms of VAL_CMC and CC_CMC gels are presented in Figure 5. The determined changes in drug content in the gels during 35 days of storage are presented in Figure 6 (CV percentage was less than 3%). The short-term stability studies confirmed satisfying the chemical stability of VAL in all gels during storage, either at 25 °C or 4 °C. Surprisingly, a significant decrease in CC content was observed in all investigated hydrogels stored for longer than 14 days. The change was not related either to the temperature of storage or to the type of carrier. In addition, analysis of the chromatograms did not reveal any degradation products (Figure 5). Because the loss of CC measured in the gels after 21 and 35 days was around 10%, a storage time of no longer than 14 days may be recommended for CC preparations. The reason for the lower results may be the interaction of CC with the polymer, preventing complete recovery during sample preparation for analysis. During storage, stronger adsorption of the polymer on the CC particles could have occurred, and extraction with methanol, leading to precipitation of the polymer at the interface, could have resulted in the encapsulation of some of the solid CC particles and their poor dissolution. Thus, the same extraction method of CC from the samples at later time points was inefficient. Because the results of the determination of CC in the residue after centrifugation of the methanol solution were ambiguous, a change of the extraction solvent or sonication of the samples might be a suitable solution for this analytical problem. Ultimately, taking into account the observed changes in consistency, especially in the case of the CMC medium, we cannot recommend a storage time of CC gels longer than 14 days.

## 4. Conclusions

Semi-solid oral formulations appear to be a promising extemporaneous pediatric dosage form, and they were characterized as carriers for two hypotensive drugs used off-label in pediatrics: VAL and CC, which are available in tablets indicated for adults. Preferred mechanical and rheological properties eliminate the risk of spillage and ensure homogeneity of suspended particles during storage.

The three polymers tested, HEC, CMC, and CAR, can be used as bases for oral hydrogels, with sugars improving palatability and preservatives ensuring microbial quality. Despite the similar consistency assessed organoleptically, the gels differed in viscosity and rheological properties, but this did not matter during application, and the addition of powdered tablets did not significantly change these properties. According to USP pharmacopeia [73], the beyond-use-date of preserved water containing compounded preparations is 35 days, and during such period, the physicochemical stability of the obtained VAL gels was confirmed, but due to the lack of recovery of the entire CC dose during quantitative analysis of gels stored longer, the recommended storage time for CC preparations is 14 days.

Because of the possible interaction of the active substance with the polymer or other excipients, the use of the proposed oral gels as universal bases for pharmaceutical substances suspended in the form of powdered tablets should be assessed individually, depending on the specific commercial tablets. Since all the studied properties of gels with VAL and CC were similar for all the hydrogel media used, only on the basis of observation of organoleptic properties was the CAR gel considered the most advantageous.

CAR gel has one important feature that can be taken into account—at an acidic pH, such as that in the stomach, the gel transforms into a sol, which should be beneficial for the release and absorption of the active substance. Reduced drug release may be a concern for HEC and CMC gels, especially for drug substances belonging to class II or IV BCS. However, there is a lack of research on this topic, although the common mixing of pediatric drugs with highly viscous food also poses a risk of reduced bioavailability.

## Figures and Tables

**Figure 1 pharmaceutics-16-01229-f001:**
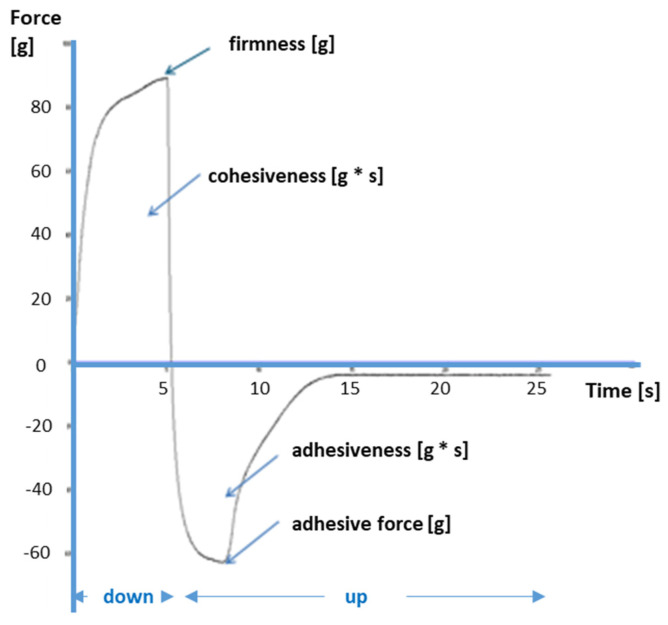
The curve and mechanical parameters of hydrogels were tested in a texture analyzer with a downstream and upstream movement of the probe.

**Figure 2 pharmaceutics-16-01229-f002:**
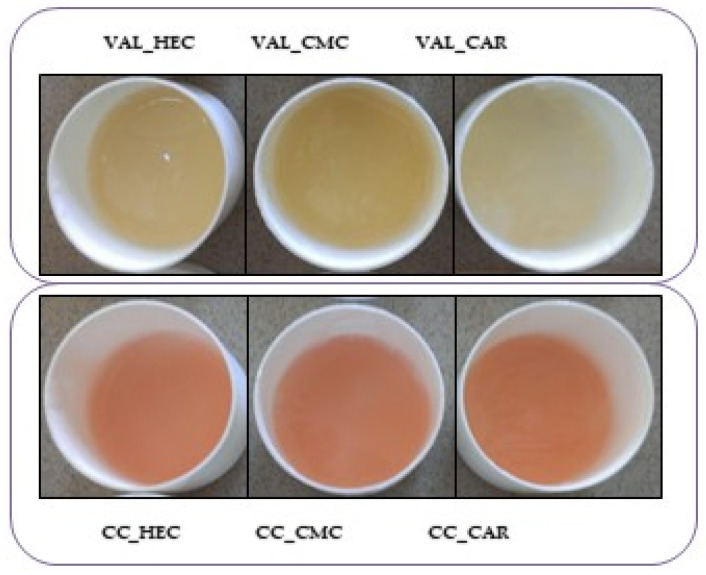
Hydrogels are prepared from valsartan (VAL) and candesartan cilexetil (CC) tablets using hydrogel carriers with hydroxyethylcellulose (HEC), carmellose sodium (CMC), and carbomer (CAR).

**Figure 3 pharmaceutics-16-01229-f003:**
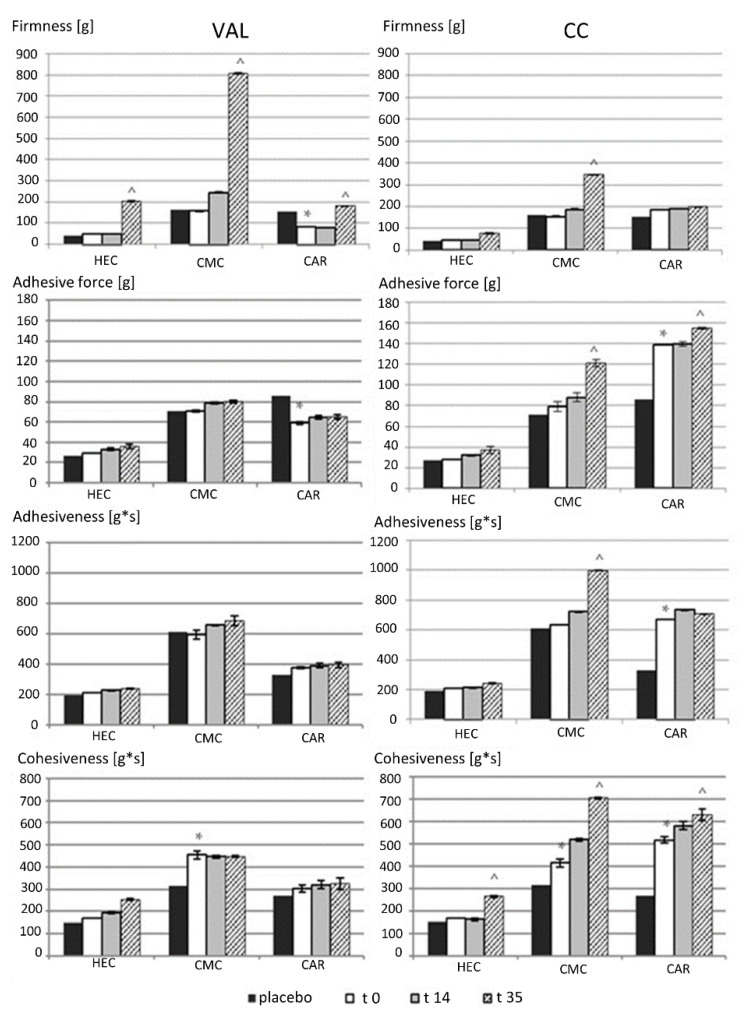
The mechanical properties of hydrogels prepared with VAL and CC tablets using hydrogel media (HEC, CMC, and CAR) and stored at 25 °C for 35 days. The mechanical parameters at t0 for the placebo hydrogels are also presented. * Statistical significance was demonstrated after the addition of the tablet mass (*p* < 0.05). ^ Statistical significance was demonstrated in relation to time t0, (*p* < 0.05).

**Figure 4 pharmaceutics-16-01229-f004:**
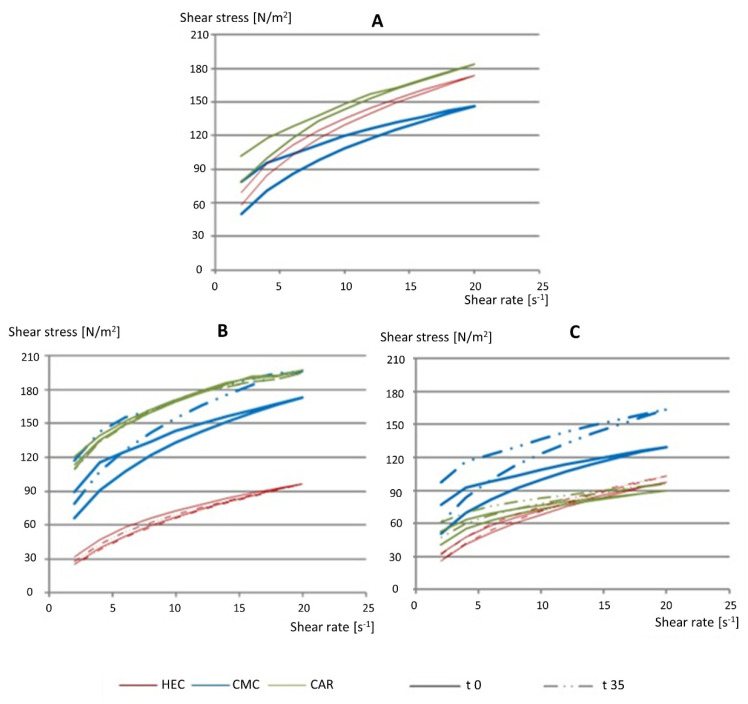
The representative hysteresis loops for HEC, CMC, and CAR gels were recorded for placebo gels at t0 (**A**) and gels with candesartan cilexetil (**B**) and valsartan (**C**) tablets after preparation (t0) and 35 days of storage at 25 °C.

**Figure 5 pharmaceutics-16-01229-f005:**
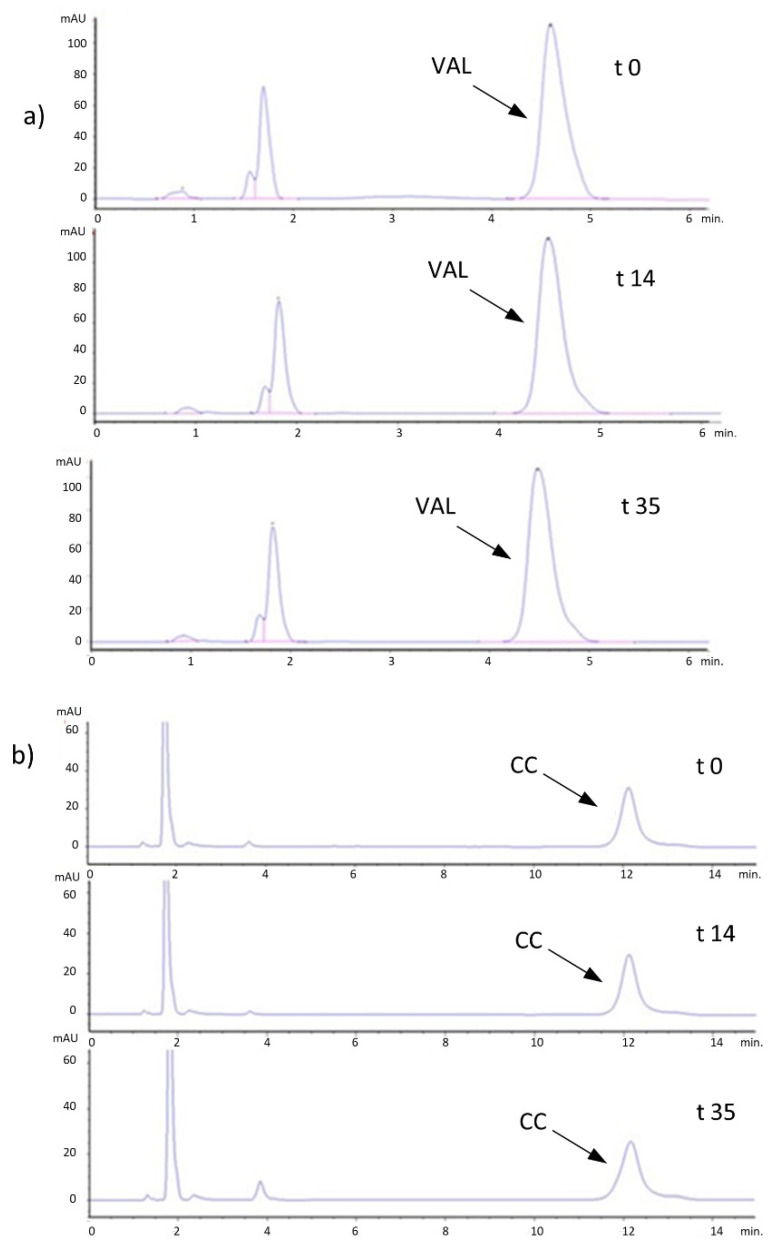
Chromatograms of VAL (**a**) and CC (**b**) in CMC gels: after preparation (t0) and after 14 and 35 days of storage at 25 °C.

**Figure 6 pharmaceutics-16-01229-f006:**
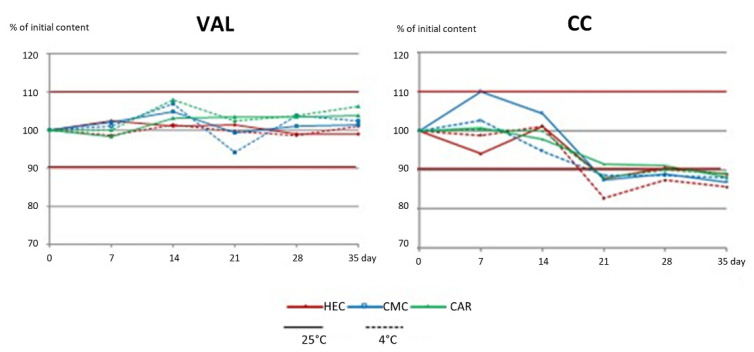
Stability (% of the initial content) of VAL and CC in oral hydrogels prepared with HEC, CMC, and CAR, stored at 25 °C and 4 °C for 35 days.

**Table 1 pharmaceutics-16-01229-t001:** Physicochemical characteristics of candesartan cilexetil (CC) and valsartan (VAL) [32,33,34,35].

Parameter	CC	VAL
Water solubility (25 °C)	pH 1.2—0.037 mg/mL pH 7.4—0.126 mg/mL	Water—0.18 mg/mL pH 8—16.8 mg/mL
pK_a_	5.9	4.7
logP	6.1	1.5
BCS classification	II	II or III

**Table 2 pharmaceutics-16-01229-t002:** Composition of placebo gels (% *w*/*w*).

Excipient	HEC	CMC (Sodium Salt)	CAR
Polymer	2.0	2.0	0.75
Sucrose	10.0		
Glycerol	5.0		
Sorbitol	4.0		
Potassium sorbate	0.10		
Methylparaben	0.10		
Water	to 100.0		

**Table 3 pharmaceutics-16-01229-t003:** Changes in viscosity (measured at a shear rate 8 s^−1^) of HEC, CMC, and CAR gels prepared with VAL and CC, stored for 35 days at 25 °C.

Gel	API	t = 0 [mPas]	t = 35 Days [mPas]	Viscosity Change [%]
HEC	---	8100	-----	
	VAL	8186	11,056	135
	CC	8244	7781	94
CMC	---	15,900	-----	
	VAL	12,907	16,173	125
	CC	16,785	20,278	121
CAR	---	9900	------	
	VAL	9178	9939	108
	CC	10,291	10,377	101

## Data Availability

The original contributions presented in the study are included in the article; further inquiries can be directed to the corresponding author/s.

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
