# Peer review of "Oral Gels as an Alternative to Liquid Pediatric Suspensions Compounded from Commercial Tablets"

_pharmaceutics, 2024, doi:10.3390/pharmaceutics16091229_

Round 1

Reviewer 1 Report

Comments and Suggestions for Authors

In the present research article (Oral gels as an alternative to liquid pediatric suspensions compounded from commercial tablets), the authors provide a comprehensive study with an appropriate design. However, there are several issues that need to be addressed before acceptance for publication. 

1.     Line 20: please define abbreviations (VAL and CC) at first time of mention

2.     Line 23: only small changes in viscosity and consistency were observed.

Please make a statistical analysis of this result and mention whether it is significant or not. Please be more scientific.

3.     Line 136 - 137: Please mention the molarity or normality of sodium hydroxide solution.

4.     Line 218 (Section 3.1): Please provide images for the prepared formulation to show their colors.

5.     Line 242 (Section 3.2): Please present data in the table as mean ± SD or a Figure showing SD values. In addition, statistical analysis will be performed to confirm the presence or absence of a significant effect of pH before and after the drug's addition.

6.     Line 414 (Section 3.4): Please show Val's chromatograms. In addition, please mention the extraction condition. maybe you did not use the appropriate organic solvent (such as acetonitrile) to dissolve polymers and ensure drug extraction. Also, did you make sonication if the organic solvent is used? Please mention all conditions to give a clear idea about decrease in CC peak.

7.     Discussion in the manuscript is very poor. Please compare your findings with those of other studies and cite them for similar research articles. 

8.  Please remove all results from the conclusion section. Provide the significance of this work and what further work could be suggested for readers. 

Reviewer 2 Report

Comments and Suggestions for Authors

The authors provided an interesting paper on the extemporal preparation of paediatric formulations of two antihypertensive drugs (valsartan and candesartan cilexetil) in the form of oral gels as an alternative to liquid formulations.

This work aimed to propose hydrogels as an alternative to suspensions. For this purpose, appropriate gel-forming polymers and methods were selected. The manuscript is logically structured and well-written.

This topic and this particular manuscript are quite important and can have a direct impact on the extemporal preparation of particular drugs in the form of gels for the paediatric population.

Proposed minor changes that are aimed at improving the manuscript are provided below:

Abstract

·       Please mention here the full names of the drugs (valsartan and candesartan cilexetil), along with their abbreviations.

Introduction

·       Please consider the following literature for covering the state-of-the-art for age-appropriate dosage-forms: 10.1016/j.xphs.2020.04.018; 10.1208/s12249-019-1534-5; 10.1080/03639045.2017.1323910;

·       Right after the description of drug products please characterise APIs.

o   It will be good mention here BCS classification here

o   pKa, LogP, and solubility incl. refs.

·       Line 100: typo “Ora”

Materials and Methods

·       Line 104: Ashland?

·       Drug products: Please provide the manufacturing site information – city & country.

·       Line 191: typo “calibration curves”

Results and discussion

·       Please discuss the effect of polyols on drug absorption compared with other dosage forms (such as syrups).

·       Please provide the physicochemical properties of selected polymers (including pKa and molecular weight) with appropriate refs.

·       Please discuss: possible physicochemical interaction between drugs, excipient’s Ca2+ from one side and polymer Carbopol from another one; and it’s effect on the rheology (FYI:www.researchgate.net/publication/298286557_Trimetazidine-Carbopol_Interaction_in_the_Matrix_Tablet/citations 

·       Please discuss the applicability of this dosage form for other BCS classes.

Round 2

Reviewer 1 Report

Comments and Suggestions for Authors

Authors address comments and manuscript can be published now.